# An Inverse Node Graph-Based Method for the Urban Scene Segmentation of 3D Point Clouds

**Bufan Zhao** [1,2], **Xianghong Hua** [1,*], **Kegen Yu** [3], **Xiaoxing He** [4], **Weixing Xue** [5], **Qiqi Li** [1], **Hanwen Qi** [1,6], **Lujie Zou** [1] **and Cheng Li** [1]

1. School of Geodesy and Geomatics, Wuhan University, Wuhan 430079, China; bufan_zhao@whu.edu.cn (B.Z.); 2015301610152@whu.edu.cn (Q.L.); hanwenqi@whu.edu.cn (H.Q.); 2015301610030@whu.edu.cn (L.Z.); lc.whu@whu.edu.cn (C.L.)
2. Key Laboratory for Digital Land and Resources of Jiangxi Province, East China University of Technology, Nanchang 330013, China
3. School of Environmental Science and Spatial Informatics, China University of Mining and Technology, Xuzhou 221116, China; kegen.yu@ieee.org
4. School of Civil and Surveying & Mapping Engineering, Jiangxi University of Science and Technology, Ganzhou 341000, China; hexiaoxing@whu.edu.cn
5. Department of Shenzhen Key Laboratory of Spatial Smart Sensing and Services, Shenzhen University, Shenzhen 518060, China; weixingxue@whu.edu.cn
6. State Key Laboratory of Information Engineering in Surveying, Mapping and Remote Sensing, Wuhan University, Wuhan 430079, China
* Correspondence: xhhua@sgg.whu.edu.cn

**Abstract:** Urban object segmentation and classification tasks are critical data processing steps in scene understanding, intelligent vehicles and 3D high-precision maps. Semantic segmentation of 3D point clouds is the foundational step in object recognition. To identify the intersecting objects and improve the accuracy of classification, this paper proposes a segment-based classification method for 3D point clouds. This method firstly divides points into multi-scale supervoxels and groups them by proposed inverse node graph (IN-Graph) construction, which does not need to define prior information about the node, it divides supervoxels by judging the connection state of edges between them. This method reaches minimum global energy by graph cutting, obtains the structural segments as completely as possible, and retains boundaries at the same time. Then, the random forest classifier is utilized for supervised classification. To deal with the mislabeling of scattered fragments, higher-order CRF with small-label cluster optimization is proposed to refine the classification results. Experiments were carried out on mobile laser scan (MLS) point dataset and terrestrial laser scan (TLS) points dataset, and the results show that overall accuracies of 97.57% and 96.39% were obtained in the two datasets. The boundaries of objects were retained well, and the method achieved a good result in the classification of cars and motorcycles. More experimental analyses have verified the advantages of the proposed method and proved the practicability and versatility of the method.

**Keywords:** point cloud semantic segmentation; construction of graph; graph cut; higher-order CRF optimization

## 1. Introduction

With the development of sensing devices and urbanization, smart cities have become an important concept in urban understanding and management. Three-dimensional laser scanning is a fast and accurate technique to obtain high-precision 3D geographical distributions of urban scenes [1–3]. Three-dimensional point cloud data have become a very useful data source in perceiving urban elements. Parameterization of 3D point clouds is an important way to reconstruct urban scenes, which refers to the transformation of basic 3D data into parameters that represent objects, and provides reliable materials for the 3D reconstruction of cities. The semantic segmentation of 3D point clouds is a fundamental

process for their parameterization and allocates the category parameter to each point [4]; thus, the computer knows 'who' the point cloud is, and can decide how to deal with them next.

Due to massive points obtained from urban scene, the computer needs to reorganize point clouds into primitives with similar properties, in order to save costs and group homologous points. The voxelization technique divides raw data into regular patches; the authors in [5] proposed an initial simplification of point clouds based on regular voxelization of the space, whereas in [6], voxels are coded in their rank numbers to explore the topological relationship between primitives. Similar to super-pixel, supervoxels are an extension of voxelization, which generates irregular patches based on rules. To deal with different geometric feature extraction, many methods generate super-voxels by a particular order of rules [7,8]. After primitive reorganization, different strategies are designed to group primitives into segmented objects [9]. Region growth is a common way to aggregate primitives which belong to the same object. Yang et al. over-segmented point clouds into supervoxels, then merged the adjacent supervoxels based on Euclidean distance into units by encoding the semantic knowledge as the merging rules [10]. Graph clustering methods such as graph cuts, including normalized cuts [11] and min cuts [8], provide a feasible and effective way to aggregate primitives with similar properties. However, there is a premise of segmentation by the graph method; it needs to know to which category or which segment the node belongs. Then, global optimization is carried out according to its suitability with the category or segment, and this prior information normally needs to be defined and obtained in advance.

Segmentation provides the sets of points which belong to an individual object; the next step is to recognize the category of the point sets. Knowledge-based methods and machine-learning-based methods are two effective classification strategies. Knowledge-based methods combine contextual information and geometric features. Based on the context knowledge of urban objects, Yang proposed formal representations of seven common types to merge supervoxels [10]. An approach to the rough classification of mobile laser scanning data based on raster image processing techniques was presented by Hůlková [12], which is based on the specified criteria (e.g., in height, size, position) of the objects in the raster image to judge their categories. Another knowledge-based method is to design shape descriptors to find the specific category, which is widely used in the classification and registration process of point clouds [13,14]. A pairwise 3D shape context descriptor was proposed to accurately extract street light poles from nonground points, which was capable of performing partial object matching and retrieval [15]. Compared to artificial rules, the machine learning method provides a more intelligent strategy for the classification of urban objects. Supervised classifiers are designed in machine learning methods and a set of features or explanatory variables are used to establish relationships between primitives and categories. Yang and Dong calculated multiple aggregation levels of contextual features; then, a C-SVM (support vector machine) was trained on the manually labeled training datasets, which used the radial basis function (RBF) kernel [16]. Niemeyer presented a contextual classification methodology based on a two-stage conditional random field (CRF), where both interaction potentials of point-based CRF and segment-based CRF were designed by random forest (RF) classifiers [17]. Lately, the 3D point cloud deep learning method has become a current research hot spot; PointNet [18] and its derived methods [19–21] provide end-to-end deep learning methods to enable high-accuracy 3D object recognition and object detection in instance segmentation. The three-dimensional point cloud deep learning method provides a feasible way for realizing the automated extraction of urban elements.

Combining the segmentation and classification methods described above yields the collective semantic segmentation methods [22]. From classification units, these can be categorized as pointwise, voxel-based and segment-based methods. Weinmann proposed a representative pointwise classification method [23–25], which provided a systematic step for supervised semantic segmentation, and inspired a lot of subsequent works [25]. Luo

proposed a supervoxel-based method to classify objects beside roads in city scenes [26], which used an idea of active learning to train classifiers, and a higher-order CRF was used to refine the category existing in the clique. A contextual segment-based classification of airborne laser scanner data was proposed by George Vosselman, which firstly segmented points into segments using parametric and non-parametric approaches, and then carried out the classification process using segments as classification units [27]. Each unit-based method has its advantages and disadvantages, which will be discussed later. In the physical world, all objects stand based on the ground, and ideally, when eliminating the ground points by some useful ground filters, the non-ground object points can gather as an integrated object and are more easily classified. One difficult problem is that the non-ground objects are sometimes next to each other or intersecting with each other; therefore, it is hard to divide and classify them correctly as well as to extract the complete point cloud of an object. To prevent under-segmentation, the classification units are usually smaller than the structure of the object in the classification process; thus, the classification of the points needs to be refined in the trivial parts to retain the category consistency in an object. To alleviate these problems, this paper proposes a segment-based classification method, the purpose of which is to obtain over-segmentation segments which are as close as possible to the complete structure of the object, and at the same time, to retain the boundary between the intersecting objects. Thus, the proposed method retains more shape features of the object and avoids confusing the classification of adjacent objects. Using higher-order CRF, this paper aimed to further optimize the errors of the trivial part. The main contributions of this work are summarized as follows:

1.  Structural segmentation of point clouds with an inverse node graph. In unsupervised segmentation steps, in order to solve the problem that the node of the graph needs prior information or definition, this paper proposes an inverse node graph which focuses on the state of edges in the graph rather than nodes, taking the connection state of the edge as the solution of the graph. It divides points into groups by judging the connection between the units and retains boundaries between intersecting objects by the inverse label operation of non-connection edges. Due to these two inverse operations, this graph is referred to as the inverse node graph (IN-Graph) in this paper. Thus, it can obtain globally optimal over-segmentation structural segments, which retain more shape feature and boundary information of objects, and provide meaningful segmentation units for the classification step;

2.  Label refinement via higher-order CRF considering small-label cluster optimization. In classification optimization steps, in order to correct the mislabeling of the scattered fragments inside a clique, this paper deals with small clusters with the same label individually based on the idea of higher-order CRF. To reduce the computational complexity, a simplified algorithm is proposed to solve the problem of small-label cluster optimization to improve the efficiency, and the classification results are further optimized.

The remainder of this article is organized as follows. Section 2 describes the details of the methodology of the proposed method. Section 3 presents the experimental results of two urban datasets, performance comparisons and discussion, and the concluding remarks are given in Section 4.

## 2. Materials and Methods

### 2.1. Overall Flow of Method

The proposed method only needs the position coordinates x, y, z in the raw point clouds and does not rely on additional data such as RGB or intensity. The overall flow of the proposed semantic segmentation method is shown in Figure 1, which includes four steps: point cloud pretreatment, unsupervised segmentation, supervised classification and label refinement.

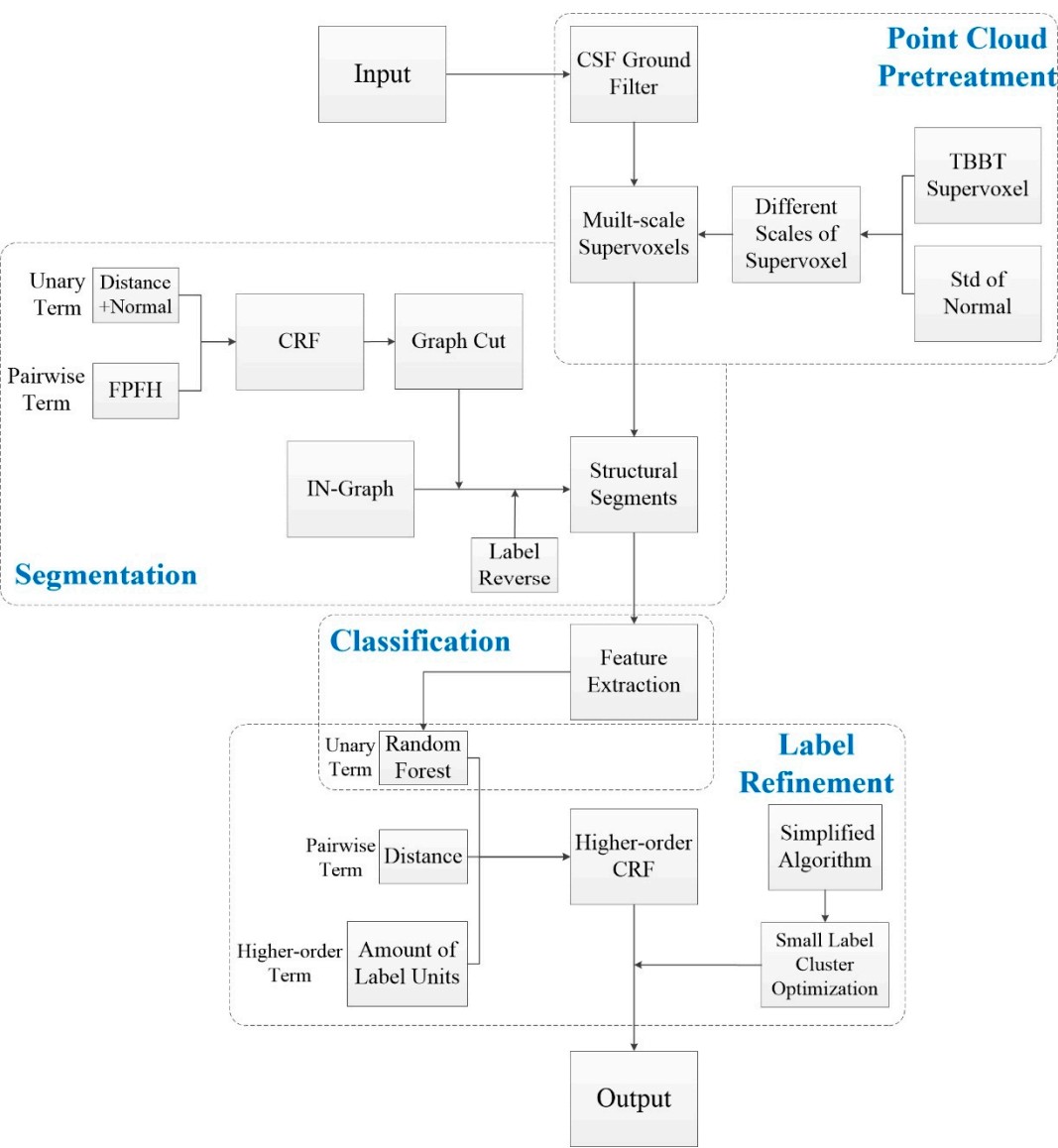

**Figure 1.** Overall flow of the proposed semantic segmentation method.

For point cloud pretreatment, the ground filter is used to extract the ground point cloud to release the influence when extracting the structure segments of objects on the ground. The non-ground point clouds are divided into different scales of voxels by the multi-scale supervoxel method, in which supervoxel generation greatly reduces the amount of input data in the subsequent process. Then, the proposed IN-Graph is used to build the graph between multi-scale supervoxels and generate over-segmentation structure segments as the basic unit of classification by graph cuts in the segmentation step. The preliminary classification labels of the structure segments are obtained by the random forest classifier in the supervised classification step, and a higher-order CRF is taken as the global optimization to refine classification results by taking small-label clusters into account. The details of the method are described step-by-step in the remainder of this section.

### 2.2. Point Cloud Pretreatment

The purpose of the pretreatment step in this paper is to reduce the input data and optimize the processing units. The main processes include the extraction of ground point clouds and the generation of multi-scale supervoxels.

### 2.2.1. Ground Point Clouds Extraction

Considering that objects such as trees, building and cars are supported and connected by the ground in real scenarios, the ground points are removed first in order to retain the independence of objects from the ground, and at the same time, it can reduce the amount of raw data [28]. The cloth simulation filtering (CSF) method is widely used in ground point cloud extraction [29], which simulates a cloth falling and sticking to the ground; the simulation of the surface greatly reduces the number of control parameters. Considering the large span of the urban scene and ensuring the accuracy of ground filters, the ground points were extracted in subsections by CSF in this study.

### 2.2.2. Multi-Scale Supervoxel Generation

After the raw point clouds are divided into ground points and non-ground points by the ground filter, the next step is to segment the non-ground points into structure segments. To aggregate the structural segment points of objects well, supervoxel technology is first adopted to cluster the points according to similar spatial and geometrical properties. Compared with the VCCS method [30], the TBBP method generates supervoxels which retain more boundary information of objects [31]; therefore, it is taken as basic voxelization method in this paper. However, the resolution of supervoxels will affect the accuracy of segmentation—an overly high resolution will produce an under-segmentation supervoxel and confuse the boundary between two adjacent objects, whereas a resolution which is too low will bring a lot of computation, and the advantages of voxelization will be insignificant.

In this paper, an adapting resolution of supervoxels is applied to generate multi-scale supervoxels [7]. Man-made objects in a city scene usually have similar geometrical features; for example, buildings and cars, which are the most common man-made objects, have different scale planar features on their surfaces. The planar dimensional feature is used to discriminate whether the supervoxel needs further subdivision.

For each supervoxel group $S_i = \{p_1, p_2, \ldots, p_n\}$, $p_i = (x_i, y_i, z_i)$, the centroid $C_i = \frac{1}{n} \sum_{j=1}^{n} p_j$ is calculated as the center of the supervoxel, and then the principal component analysis (PCA) method is performed on $C_i$ and the points to obtain the 3D structure tensors with three eigenvalues, $\lambda_1$, $\lambda_2$, and $\lambda_3$ ($\lambda_1 > \lambda_2 > \lambda_3$). The dimensional feature of linearity $L_\lambda$, planarity $P_\lambda$ and scattering $S_\lambda$ features are obtained in Equation (1) [32]. If $P_\lambda > L_\lambda$ and $P_\lambda > S_\lambda$, which means the planarity of the supervoxel is more significant, then the current supervoxel is more likely to be part of an entire object and no more subdivisions are needed.

$$L_\lambda = \frac{\sqrt{\lambda_1} - \sqrt{\lambda_2}}{\sqrt{\lambda_1}} \quad P_\lambda = \frac{\sqrt{\lambda_2} - \sqrt{\lambda_3}}{\sqrt{\lambda_1}} \quad S_\lambda = \frac{\sqrt{\lambda_3}}{\sqrt{\lambda_1}} \tag{1}$$

The supervoxel of planarity with a significantly large resolution may include some non-planarity points; therefore, a normal vector constraint is imposed to restrict the resolution of the supervoxel. If the normal vectors of points in a supervoxel are divergent, it means that the points in the supervoxel have directional discrepancy and the supervoxel needs to be subdivided. The directional discrepancy is calculated as the standard deviation of the angles between the normal vectors of points and the normal of the supervoxel as follows:

$$N_i = \frac{1}{m} \sum_{j=1}^{m} n_j \tag{2}$$

$$df_i = \operatorname*{std}_{j=1\ldots m} \left( |1 - N_i \cdot n_j| \right) \tag{3}$$

where $n_j$ is the normal vector of point $p_j$ in $S_i$ and $N_i$ is the normal vector of $S_i$, $m$ is the total number of points in $S_i$, and $df_i$ is the directional discrepancy of the supervoxel. Combining the planarity and the directional discrepancy, the criteria for supervoxel subdivision can be expressed as:

$$\begin{cases} retain & if\ P_\lambda > L_\lambda \bigcap P_\lambda > S_\lambda \bigcap df < \Delta df \\ subdivide & or\ else \end{cases} \tag{4}$$

where $\Delta df$ is an adjustable parameter to weight the dispersion of points in the supervoxel. Algorithm 1 gives a description of the steps of the generation of a supervoxel of three scales in this paper.

---

**Algorithm 1.** Multi-scale supervoxel generation

---

Input: point clouds $P$, selectable resolution R = $(r_1, r_2, r_3)$ where $r_1 > r_2 > r_3$
Output: Multi-scale supervoxel set S
1: Initialize input points $P_s = P$
2: for $r_i = r_1, r_2$ do
3:     voxelization $P_s$ by the TBBT method with resolution $r = r_i$ to generate superxovel group $S_v$
4:     compute $L_\lambda$, $P_\lambda$, $S_\lambda$ of each supervoxel $s$ in $S_v$ with Equation (1)
5:     compute $df$ of each supervoxel $s$ in $S_v$ with Equation (3)
6:     for all $s$ in $S_v$ do
7:         decide whether $s$ is retained or subdivided with Equation (4): if $s$ is retained, perform $s \in S$; otherwise, $s \in \hat{S}$.
8:     end for
9:     update $P_s$ as the points in $\hat{S}$
10: end for
11: voxelization $P_s$ with resolution $r = r_3$ to generate superxovels group $S_v$, add all of supervoxels in $S_v$ in S
12: output final multi-scale supervoxel group $S$

---

*2.3. Segmentation of the Unsupervised Structural Segments via the Proposed Inverse Node Graph Construction*

Through the steps in Section 2.2, raw point clouds are transformed into multi-scale supervoxels, and the basic unit has been transformed to supervoxel-based from pointwise. Although the supervoxel-based unit contains more meaningful local feature information than the pointwise unit, the segment-based unit can retain more geometrical features of structures and has a wider range of context knowledge of objects. To restore the surface structure of objects as much as possible, supervoxels need to be aggregated into structural segments. In order to not confuse the boundary, the structural segments need to be over-segmented, and at the same time, they have to preserve enough geometrical features of objects for better decision-making by the training classifier. For this purpose, this paper generates the structural segments by constructing the proposed inverse node graph with supervoxels, and the details of IN-Graph are described below.

2.3.1. Regular Graph of Segmentation

Graph cuts are a very useful and popular global energy optimization method in point cloud segmentation. The first step is to build a graph of supervoxels, and then to cluster the supervoxels by determining to which segments they belong. The graph of supervoxels is built as $G = \{V, E\}$, where $V$ represents the nodes of graph; here, it takes the centroid of the supervoxel as representative of $V$; $E$ represents the edges between the adjacent nodes, where it constructs the Delaunary triangulation of centroids to build adjacent systems of graphs, and a distance threshold $\Delta d$ is set to restrict the maximum distance between the points in adjacent supervoxels to avoid false edges. Figure 2a shows a traditional graph, where $y$ is the category configuration of the nodes (red circle in Figure 2a).

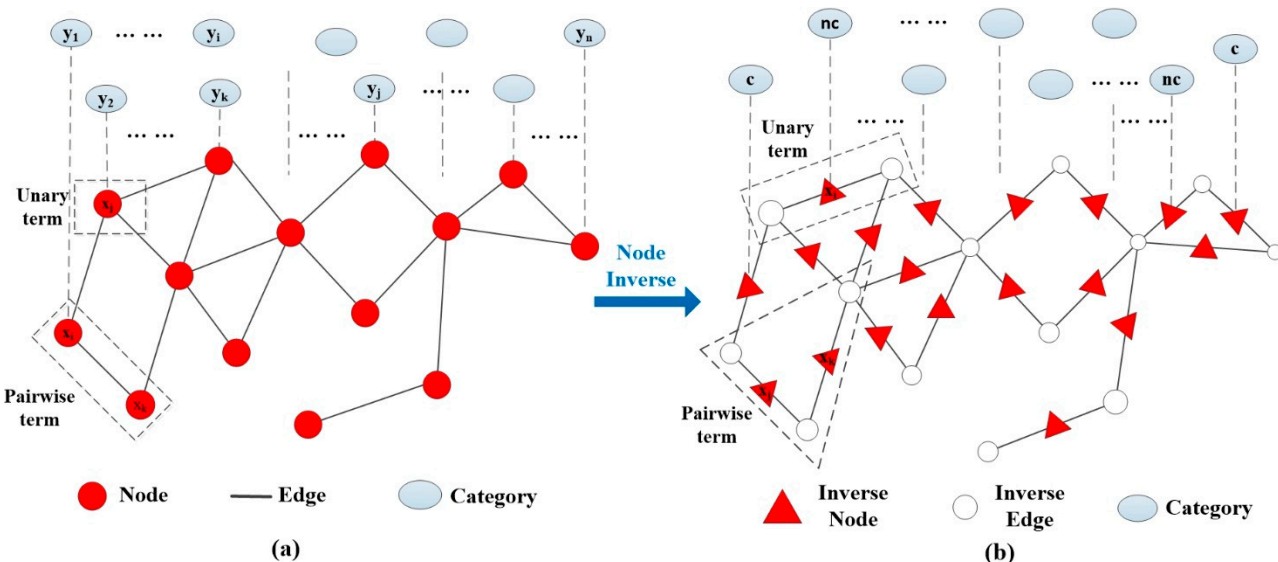

**Figure 2.** Depictions of regular graph and inverse node graph. (**a**) Regular graph. (**b**) Inverse node graph. Where all nodes and corresponding lables do not list all in figure, using the ellipsis instead.

Then, CRF can be built to solve the minimum global energy optimization of graph, and the energy function of CRF can be expressed as:

$$\min_{y} E(y) = \sum_{i \in x} \varphi(y_i) + \alpha_1 \sum_{(i,j) \in e} \omega(y_i, y_j) \tag{5}$$

where $y$ is a label configuration of the nodes, $\varphi(y_i)$ is the unary term which measures how the node fits for label $y$, $\omega(y_i, y_j)$ is a pairwise term which is used to smooth the boundary of nodes with different labels, and $b$ is a balance parameter. Then, the energy function is approximated using graph cut optimization to obtain a globally optimal solution.

2.3.2. Inverse Node Graph Generation

Segmentation is an unsupervised process; therefore, without a priori information about the scene, the number and the type of structural segments in the graph is unknown, and the label configuration y of the nodes cannot be determined. This means that it does not know to which structural segment the node belongs or how many segments could be generated in the graph, and this problem restricts the versatility of the graph method in segmentation. Some studies have defined linearity, planarity and scattering or other geometric consistencies as segmentation criteria in advance [7,33,34]; however, the limited label configuration restricts the kinds of structural segments and is prone to causing under-segmentation. The point clouds from different objects could be divided into the same group; they will be classified as the same category at the classification step, which needs an extra process to deal with under-segmentation after the classification. Therefore, the ideal state is to obtain the over-segmented structure that preserves the boundaries between the objects while maintaining the geometrical structure as much as possible.

The label configuration of notes is unknown in the graph, but the state of edges is obviously known; either disconnected or connected. According to this basic fact, a novel graph is constructed in this paper which exchanges the status of the node and edge: the solution of the graph changes to judge the state of edge. It divides two supervoxels into the same structural segments according to decide whether the edge of two supervoxels is connected or not. The proposed graph takes edge as 'node'; therefore, it is termed as an inverse node graph and the 'node' is called the inverse node, achieving real segmentation by cutting rather than clustering. The construction details of the IN-Graph are shown in Figure 2b.

As Figure 2b indicates, the decision point goes from a regular node (the circle in Figure 2a) to an inverse node (the triangle in Figure 2b). Although Figure 2b uses an entity triangle to represent the inverse node, it has no actual positional meaning and it does not include any points, it just indicates the connection status between two supervoxels. Then, the label configuration of inverse nodes is set as two categories: connection (label *c* in Figure 2b) and non-connection (label *nc* in Figure 2b). If adjacent supervoxels are similar, the inverse node could be in the connection state; otherwise, the inverse node is in a non-connection state. After determining the connection state of all inverse nodes, the regions constitute the structural segments according to connectivity of supervoxels, and it can then achieve unsupervised segmentation with the graph method even if there is no prior target label information.

### 2.3.3. Global Optimal Segmentation via Graph Cut

Similar to traditional graphs, the IN-Graph achieves global optimal segmentation by graph cuts to reach minimum global energy, as in Equation (5), where the label configuration of inverse node is set as:

$$\begin{cases} y_t = 0, & if \ inverse \ node \ is \ connection \\ y_t = 1, & if \ inverse \ node \ is \ non-connection \end{cases} \tag{6}$$

The unary term and pairwise term of the IN-Graph must fit the properties of the edges. The connection of the inverse node is determined by the similarity of two supervoxels which are on both sides of the edge; thus, the unary term of the inverse node is fitted by two supervoxels it contains, as shown in the dotted box of Figure 2b. Here, the distance discrepancy and the normal discrepancy [35] of adjacent supervoxels are taken as the unary term of the inverse node as follows:

$$\varphi(y_t) = \begin{cases} b \cdot Dis(S_i, S_j) + (1-b) \cdot Nor(S_i, S_j), & if \ y_t = 0 \\ b \cdot [1 - Dis(S_i, S_j)] + (1-b) \cdot [1 - Nor(S_i, S_j)], & if \ y_t = 1 \end{cases} \tag{7}$$

where *b* is an adjustable parameter that balances the distance discrepancy and normal discrepancy, and the distance discrepancy $Dis(S_i, S_j)$ and normal discrepancy $Nor(S_i, S_j)$ are set as:

$$Dis(S_i, S_j) = \min_{k=1...n, l=1...m} [distance(p_k, p_l | p_k \in S_i, p_l \in S_j)] / \Delta d \tag{8}$$

$$Nor(S_i, S_j) = \cos[angle(N_i, N_j)] \tag{9}$$

where $Dis(S_i, S_j)$ indicates the minimum Euclidean distance between two points in two adjacent supervoxels, $S_i$ and $S_j$, respectively, and $\Delta d$ is set as mentioned above in Section 2.3.1 to constrain $0 \le Dis(S_i, S_j) \le 1$. $Nor(S_i, S_j)$ represents the cosine of the angle between the normal vectors of $S_i$ and $S_j$, where $0 \le Nor(S_i, S_j) \le 1$. The unary term indicates that if two adjacent supervoxels are closer to each other and the directions of normal vectors are consistent, they are more likely to be a continuous surface and possibly come from the same object, and if they are segmented into the same structural segment, then $y_t = 0$; otherwise, $y_t = 1$.

The pairwise term of the IN-Graph is similar to the unary term, but covers a larger span than the unary term. As shown in the triangle frame of Figure 2b, two adjacent inverse nodes are constituted of three supervoxels, namely, one public supervoxel and two individual supervoxels. The pairwise term smooths the label of adjacent inverse nodes, and two situations could exist in pairwise terms, which are defined as:

$$\begin{cases} |D(S_1, S_2) - D(S_1, S_3)| \to 0, & if \ y_i = y_j \\ |D(S_1, S_2) - D(S_1, S_3)| \to \infty, & if \ y_i \ne y_j \end{cases} \tag{10}$$

where $S_1$ is a public supervoxel, and $D(S_1, S_2)$ represents the difference between $S_1$ and $S_2$. When $y_i = y_j = 0$, $S_1$, $S_2$ and $S_3$ are connected, the values of $D(S_1, S_3)$ and $D(S_1, S_3)$ will be small, and $|D(S_1, S_2) - D(S_1, S_3)|$ will be an even smaller value; when $y_i = y_j = 1$, they are connected, both $D(S_1, S_3)$ and $D(S_1, S_3)$ are large, and $|D(S_1, S_2) - D(S_1, S_3)|$ remains small. When $y_i \neq y_j$, it means the one of the inverse nodes is a connection and another is a non-connection; thus, one of $D(S_i, S_j)$ will be large and another will be small, which means that the difference between $D(S_1, S_3)$ and $D(S_1, S_3)$ will be large. Here, the distance in FPFH feature space of supervoxel is defined as the metric $D(S_i, S_j)$:

$$D(S_i, S_j) = \|FPFH(S_i) - FPFH(S_j)\|_2 \tag{11}$$

The FPFH feature of the supervoxel provides measurable information for the description of geometric features in the high-dimensional hyperspace, which is invariant and robust under different sampling densities or noise levels in the neighborhood; therefore, this paper takes the difference in FPFH distance between supervoxels as the pairwise term. Then, the pairwise term $\omega(y_i, y_j)$ can be defined as:

$$\omega(y_i, y_j) = \begin{cases} |D(S_k, S_l) - D(S_k, S_t)|, & if \ y_i \neq y_j \\ 0, & if \ y_i = y_j \end{cases} \tag{12}$$

where $S_k$ and $S_l$ belong to inverse node $i$, and $S_k$ and $S_t$ belong to inverse node $j$. Then, the minimum global energy in Equation (5) with the unary term and pairwise term is approximated by graph cut, and the label allocates to each of inverse node.

Due to the smooth term, inverse nodes on the surface of the discrete object, such as vegetation, are easily smoothed by a non-connection label, which leads to non-connection clusters, as Figure 3a shows. In the physical world, the boundary of two intersecting objects could be a single surface rather than multilayered. To the non-connection cluster, the real non-connection inverse node of the boundary should appear at the edge of the cluster. Then, after graph cut, am inverse label operation is used to eliminate non-connection clusters while retaining the boundary of segments, which is expressed as:

$$\bar{y}_i = 0 \ if \ [\forall y_j = 1 | y_i = 1, \ y_j \in Nei(y_i)] \tag{13}$$

where $y_i$ is the initial label, $\bar{y}_i$ is the inverse label, and $Nei(y_i)$ is the set of adjacent inverse nodes of $y_i$. As shown in Figure 3b, there was only one segment, but with the inverse label operation, another segment was constructed. The inverse label operation transforms the label of non-connection inverse nodes whose adjacent nodes are all in non-connection. While maintaining the internal integrity of the object as much as possible, it also preserves a more appropriate boundary with the physical world.

Figure 4 shows the segmentation example of vegetation and a façade, through IN-Graph construction and inverse label operation. Supervoxels are divided into over-segmented structural segments, which restore boundary information and geometrical features. The vegetation close to the wall has been extracted by graph cut and inverse label operation, as shown in the blue box of Figure 4b,c. Next, the structural segments can be treated as basic input units in the supervised classification step.

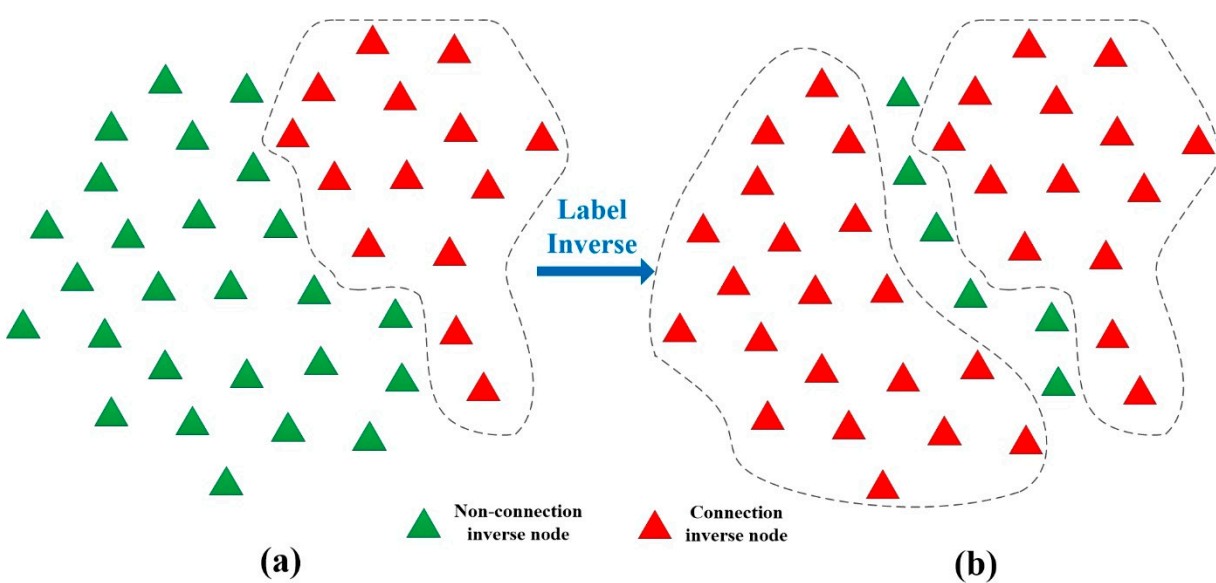

**Figure 3.** Explanation of inverse label operation. (**a**) Before inverse label operation. (**b**) After inverse label operation.

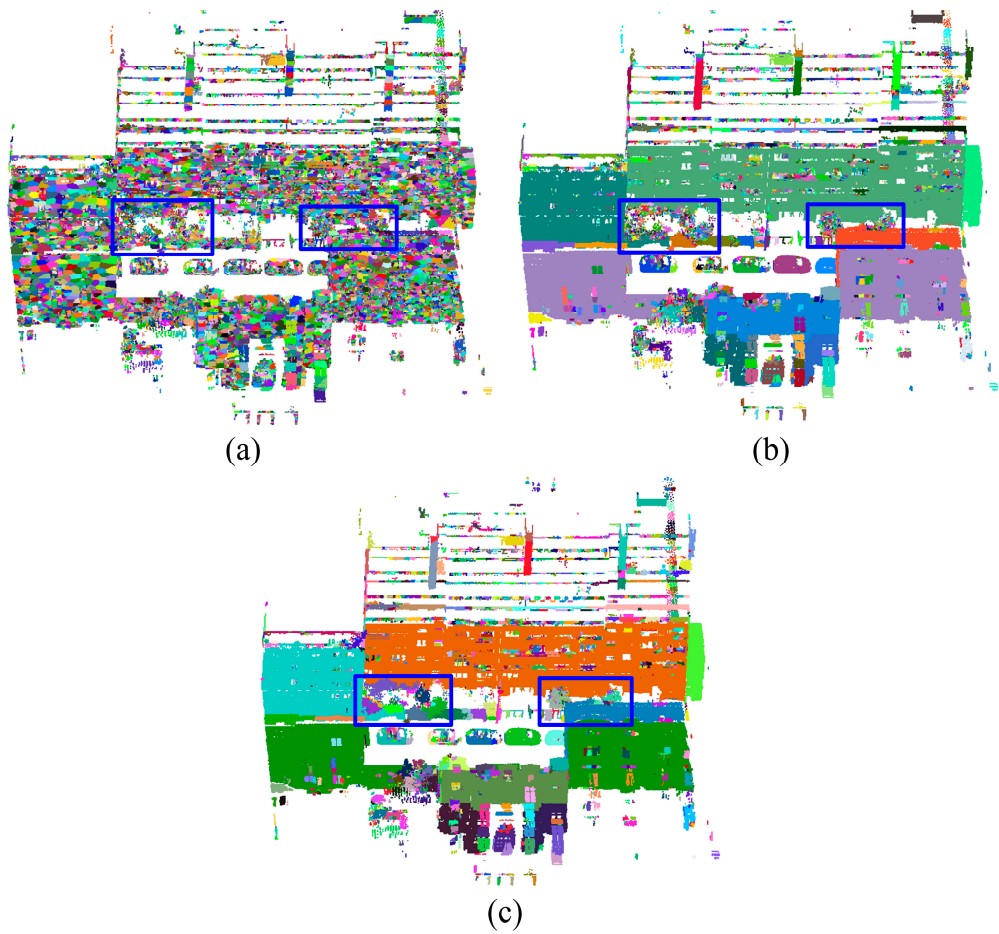

**Figure 4.** The views of each step in IN-Graph segmentation. (**a**) Multi-scale supervoxel of the point cloud. (**b**) Results of the graph cut. (**c**) Results of the inverse label operation.

*2.4. Supervised Classification via Higher-Order CRF Considering Small-Label Clusters*

After the structural segments retaining the geometrical structures and boundaries of objects are obtained, semantic labels will be allocated to these structural segments by a supervised higher-order CRF classification model. The graph model with higher-order structural segments is constructed as $G = \{E, V, C\}$, and the graph construction is the same as in Section 2.3.1, although the difference is the addition of a higher-order term of label cost in the region clique $c$. The higher-order CRF is defined as:

$$\min_{y \in labels} E(y) = \sum_{i \in V} \varphi_i(y_i) + \alpha_2 \sum_{(i,j) \in E} \omega_{i,j}(y_i, y_j) + \beta_2 \sum_{c \in C} \delta_c(y_c) \tag{14}$$

where $\alpha_2$ and $\beta_2$ are balance parameters, $\delta(y_c)$ is the higher-order term, and $C$ indicates the set of cliques. Detailed descriptions of the configuration of the unary term, pairwise term and higher-order term are given below.

2.4.1. Configuration of the Unary Term and Pairwise Term

In order to obtain the semantic label of the structural segment, a supervised classifier is trained for the initial labels of the structural segments. Here, random forest (RF) is taken as the supervised classifier, which has a good performance in point cloud classification [36], and the configuration of the input features is described in Table 1. After training and prediction, the predicted labels of probability are taken as the unary term of higher-order CRF, and the unary term is defined as follows:

$$\varphi_i(y_i) = \exp(1 - P_i) \tag{15}$$

where $P_i$ is the probability that the segment $x_i$ is allocated to label $i$ from RF, and the unary term is used to measure how the structural segments fit into the classified category.

**Table 1.** Configuration of the input features of RF in this paper.

| Features | Description | Dims |
|---|---|---|
| Mean eigenvalue-based 3D features | The mean value of eigenvalue-based 3D features and verticality of all supervoxels in the structural segment. | 9 |
| Number of linear, planar, and scatter supervoxels | The number of linear, planar and scatter supervoxels in the structural segment, and the outstanding features of them. | 4 |
| Height information | The height of the center of structural segment points from the ground; maximum and minimum heights of structural segment points from ground; the difference between the maximum and the minimum. | 4 |
| Number of supervoxels | The number of supervoxels contained in the structural segments. | 1 |
| Size of 3D and 2D minimum bounding box | Length, width, height and volume of the structural segment; 2D length, width and area of the projection of the structural segment. | 7 |
| FPFH | Mean FPFH of supervoxels in the structural segment. | 33 |
| Total | | 58 |

The pairwise term is used to punish the adjacent nodes with different labels and smooth the intersection of the structural segments. The minimum Euclidean distance between two adjacent segments is defined as the pairwise term as follows:

$$\omega_{i,j}(y_i, y_j) = \begin{cases} \exp(-dis(x_i, x_j)), & if \ y_i \neq y_j \\ 0, & if \ y_i = y_j \end{cases} \tag{16}$$

$$dis(x_i, x_j) = \min_{k=1...n, l=1...m} \left[ distance(p_k, p_l \mid p_k \in x_i, p_l \in x_j) \right] \tag{17}$$

### 2.4.2. Higher-Order Term with Small-Label Cluster Optimization

The higher-order potential has been proven to be effective in refining classification results in the regional clique [26,33,37,38]; the clique represents a region which is connected in the graph. The definition of higher-order terms in different works are similar, which constrain the multi-labels which exist in a clique, and the label term heavily penalizes the category label when there are a few units allocated to this category in the clique. The point of the higher-order term first focuses on the category, and then on the segment units. The definition of the higher-order term is defined as:

$$\delta_c(y_c) = \sum_{l \in L} h_l^c \cdot \vartheta_c(l) \tag{18}$$

$$h_l^c = \begin{cases} 1, & \exists x_i \in c \ : y_i = l \\ 0, & otherwise \end{cases} \tag{19}$$

$$\vartheta_c(l) = \begin{cases} \exp\left(\frac{M_l - |c(l)|}{M_l}\right), & |c(l)| < M_l \\ 0, & otherwise \end{cases} \tag{20}$$

where the configuration of the higher-order term is the same as that in [26]. $|c(l)|$ represents the number of units with label $l$ in a clique, because the size of structural segments in a clique could have differ greatly in different objects; here, $|c(l)|$ takes the number of supervoxels inside the clique. The empirical truncation threshold $M_l$ is defined based on the number of supervoxels inside the structure segments of each category from the training data.

If a clique has two categories, one correct category and another incorrect category, the incorrect category with fewer units is dispersive in the clique, but there are too many of them which exceed the constraint threshold value $M_l$, so that the higher-order term cannot eliminate the existence of the incorrect category. As shown in Figure 5a, there are several places where the wall fragments are mislabeled as vegetation in the clique, as marked in the green and yellow boxes in Figure 5a. Mislabeling occurs on the edge, and the number of units labeled as vegetation (include true vegetation units and incorrect vegetation units) inside this clique is large; therefore, the pairwise term and higher-order term cannot fix these mistakes. These small clusters with the same label, called the small-label cluster in this paper, need to be handled separately.

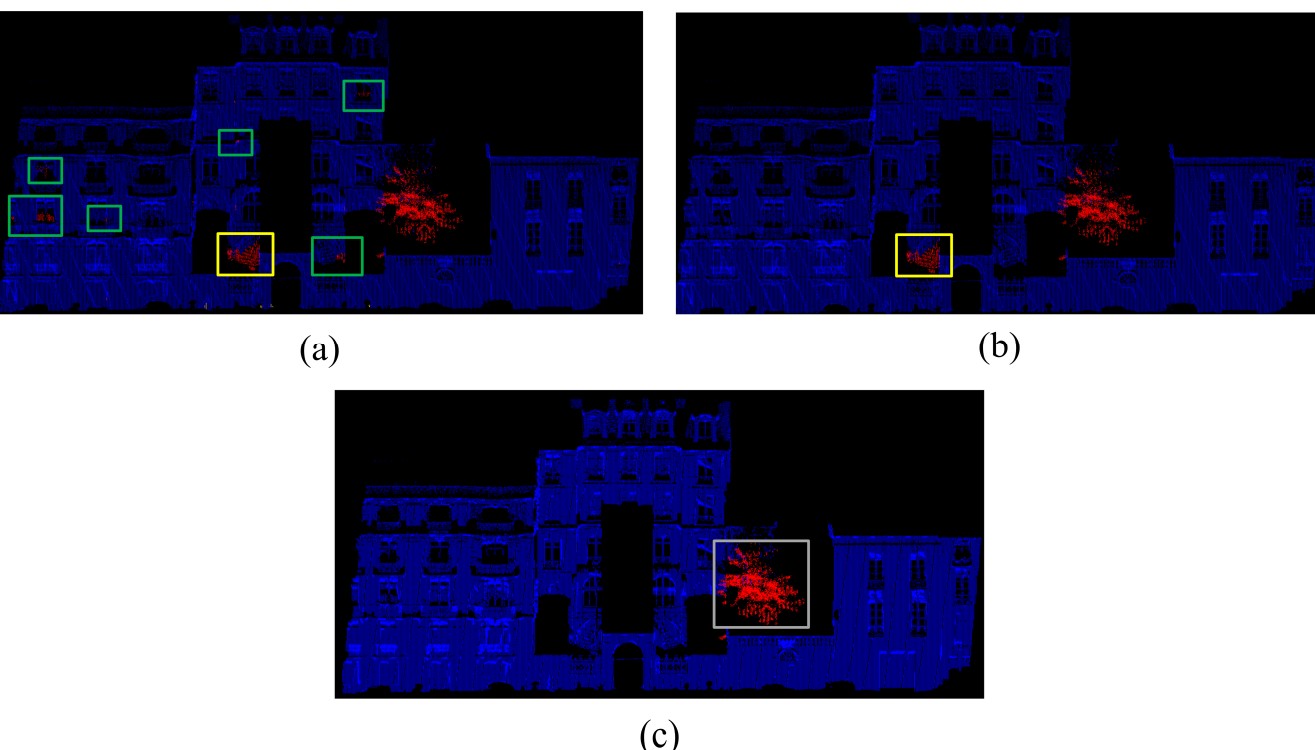

**Figure 5.** Illustration of small-label cluster optimization. (**a**) Results of a clique from higher-order CRF refinement. (**b**) Result of higher-order CRF with a small-label cluster when $M_{vegetation}$ = 50. (**c**) Result of higher-order CRF with a small-label cluster when $M_{vegetation}$ = 150.

To deal with this situation, firstly, the initial label of the segment is obtained from higher-order CRF optimization; then, it focuses on the small-label clusters in the clique, refining one of the small-label cluster with Equation (14) which ignores other small clusters with the same labels. This means that it deals with each small-label cluster independently without considering the existence of other units with the same labels in the clique, in order to avoid the total number units with such a label which need to be refined being too large. The definition of the small-label clusters is defined as:

$$mc_l(k) = \left\{ x_i, x_j \middle| \forall y_i = l, dis(x_i, x_j) < \Delta d, \quad i, j = 1 \dots n \right\} \tag{21}$$

where $mc_l(k)$ is the $k$th small clusters of the structural segments with category $l$ in clique $c$. $x_i$ and $x_j$ are the units of $mc_l(k)$. Thus, the higher-order CRF is run for all units in the clique to obtain the global optimum, in order to refine the few units in the target small-label cluster. Running higher-order CRF to refine all small-label clusters in the clique would take too long, consume too many computational resources and be wasteful. To reduce the computation complexity, this paper designs a simplified algorithm. The process focuses on the category of the target small-label cluster, which does not reallocate the label of all units in the clique and only reallocates the label of the target small-label cluster. The simplified algorithm does not need obtain global optimization because the refinement targets are only the units in the small-label cluster; therefore, it just needs to consider the local optimum for the target units. The energy function of the small-label cluster is defined in Equation (22), finding the new label of $mc_l(k)$ that can achieve the minimum $E_{mc(k)}(y)$.

$$E_{mc(k)}(y) = \sum_{\substack{i \in V \\ V \in mc_l(k)}} \varphi_i(y_i) + \alpha \sum_{\substack{(i,j) \in E \\ E \in R(mc_l(k))}} \omega_{i,j}(y_i, y_j) + \beta \sum_{sc \in c} \delta_{sc}(y_{sc}) \tag{22}$$

where $l$ is an initial label from the higher-order CRF, $R(mc_l(k))$ is the adjacent region of $mc_l(k)$ and $sc = \{c - mc_l(i) | i \neq k, i = 1 \dots n\}$. The configuration of each term is the same

as in Equation (14), although working in a local region $R(mc(k))$. More details of the simplified algorithm are described in Algorithm 2. The key point is when the total number $|sc(l)|$ of units in the clique is computed, it ignores other small-label clusters with the same initial label $l$ and assumes that only this cluster with initial label $l$ is in this clique. The refined label is allocated to each small-label cluster by Equation (22) independently; thus, it can handle the discrete mislabeling clusters in the clique.

Figure 5 illustrates how the higher-order CRF with small-label cluster optimization effectively handles the error classification clusters with different sizes in a clique. Figure 5b is an example of an optimization for Figure 5a when $M_{vegetation}$ is set to 50; it can be seen that the initial labels of some smaller clusters which are incorrectly allocated as vegetation, marked in green boxes in Figure 5a, have successfully been refined as façades. However, in the yellow boxes in Figure 5a,b, the total of units in this single label cluster is too large to fix when $M_{vegetation}$ is set as 50; when increasing $M_{vegetation}$ to 150, the small-label cluster in the yellow box has been corrected, and the real vegetation cluster is still retained, as shown in the grey box in Figure 5c.

---

**Algorithm 2.** Higher-order CRF with small-label cluster optimization

---

Input: initial label $l$ from the higher-order CRF
Output: refined label $\bar{l}$
1: define the cliques $C$ according to the connectivity of the structural segments
2: for $c$ in $C$ do
3:   for l in labels do
4:     define the small-label cluster $mc_l$ in $c$ with Equation (21)
5:     for $mc_l(i)$ in $mc_l$ do
6:       if the number of supervoxel in $mc_l(i)$ < $\Delta$n
7:         find the label set L existing in $R(mc_l(i))$
8:         for $l_{mc}$ in L do
9:           compute $E_{mc(i)}(l_{mc})$ with Equation (22).
10:        end for
11:        $\bar{l}_{mc} = l_{mc}$ where $E_{mc(i)}(l_{mc})$ is minimum.
12:      end if
13:    end for
14:  end for
15: end for
16: updata labeling $\bar{l}_{mc}$

---

## 3. Experimental Evaluations and Discussions

### 3.1. Dataset Illustration

To evaluate the performance of the proposed classification method, this study conducted the experiments on two different urban scene datasets. The first one was a benchmark MLS dataset Paris-rue-Cassette [39], which is widely used to test the performance of semantic segmentation methods, as shown in Figure 6. In the original label, some vegetation close to the wall was marked as façade, as shown in the yellow box in Figure 6b—it has been revised to vegetation, as shown in Figure 6c. The second dataset was a multi-station TLS dataset which was collected at Wuhan University, as shown in Figure 7. The descriptions of the two datasets are given in Table 2. These two datasets were collected by different platforms, so they have different qualities of the point cloud; the difference between the Wuhan University dataset and Paris-rue-Cassette dataset is that the density of the Wuhan University dataset is uniformly distributed. This study tested our semantic segmentation method with different kinds of point cloud.

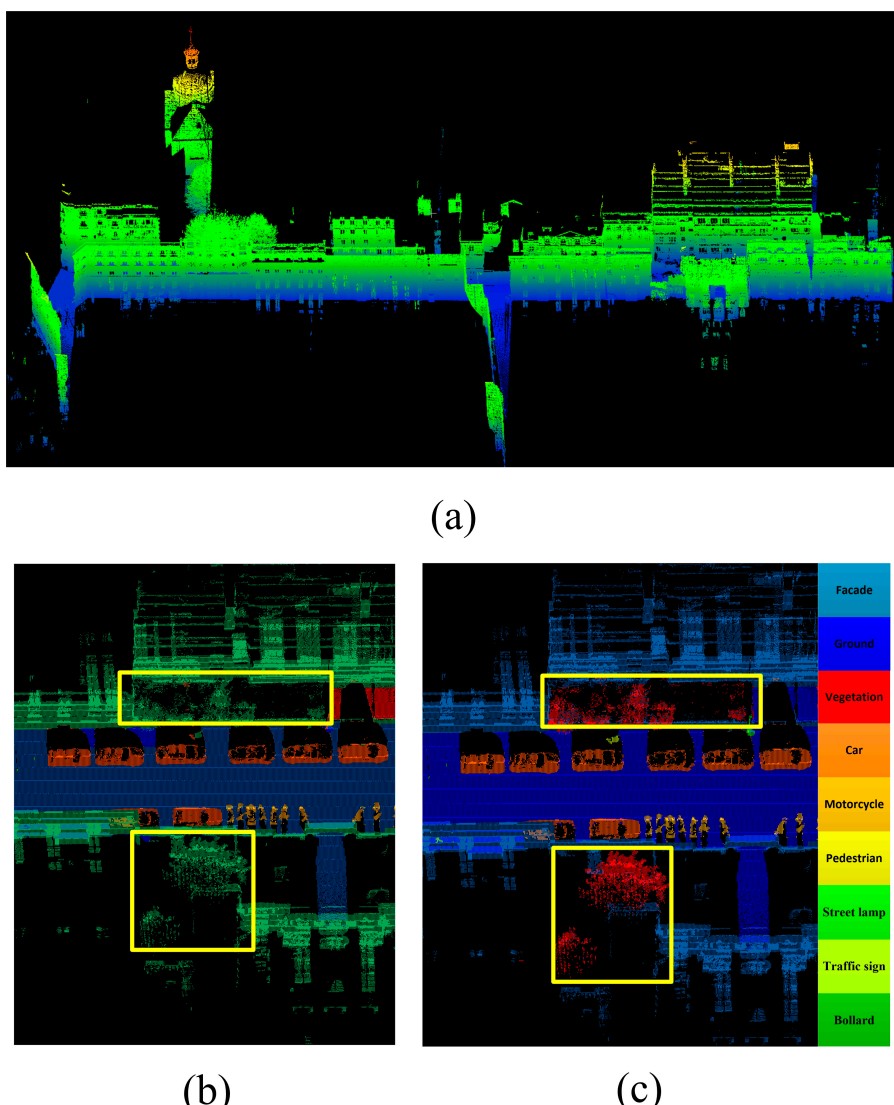

(a)

(b) (c)

**Figure 6.** Details of the Paris-rue-Cassette dataset. (**a**) The original point clouds. (**b**) The region of mislabeling points of vegetation. (**c**) Correction for mislabeling points of vegetation.

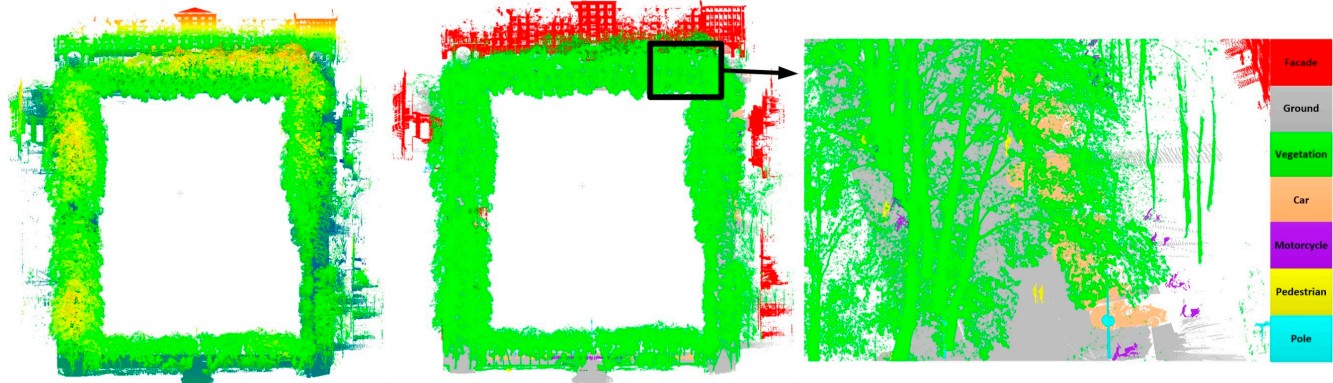

**Figure 7.** Details of the Wuhan University dataset.

**Table 2.** Characteristics of the two datasets.

| Dataset | Density (m) | Precision | Size | Quality | Category | Length (m) | Platform | Instrument |
|---|---|---|---|---|---|---|---|---|
| Paris-rue-Cassette | 0.03~0.1 | 10 m:5 mm | 12,000,000 | Medium | 9 | 200 | MLS | STEREOPOLIS II |
| Wuhan University | 0.02 | 10 m:2 mm/ 25 m:3.5 mm | 31,310,761 | High | 7 | 700 | TLS | FARO Focuss350 |

The parameter configurations of the proposed method in two datasets are shown in Table 3. It seems that many parameters need to be set, which could influence the stability of the method. In actual application, the parameters of the supervoxel can fit most situations, the distance threshold $\Delta d$ of adjacent units can depend on the density of data, and the scalar threshold $\Delta n$ and $M_l$ of each category are set as empirical values. The parameters that need to be adjusted mainly focus on the weight of distance differences and normal difference in the IN-Graph method, and the weights of the unary term, pairwise term and higher-order term in graph cut optimization.

**Table 3.** Parameter settings for two datasets.

| Step | Supervoxels | | | IN-Graph | | Higher-Order CRF with Small-Label Cluster | | |
|---|---|---|---|---|---|---|---|---|
| Parameters | $\Delta df$ | $r$ (m) | $\alpha_1$ | $b$ | $\Delta d$ (m) | $\alpha_2$ | $\beta_2$ | $\Delta n$ |
| Paris-rue-Cassette | 0.5 | 1.0/0.5/0.3 | 0.5 | 0.4 | 0.2 | 0.2 | 3 | 100 |
| Wuhan University | 0.5 | 1.0/0.5/0.3 | 0.3 | 0.3 | 0.2 | 0.1 | 3 | 100 |

### 3.2. Experiment on Paris-Rue-Cassette MLS Dataset

The Paris-rue-Cassette dataset was collected by MLS, and it includes nine categories, which are ground, façade, vegetation, car, motorcycle, pedestrian, traffic sign, street lamp and bollard. The most remarkable property of the dataset is that in the direction of the vehicle, the sampling interval is 0.1 m, and in the direction of the scanner line, the sampling interval is 0.03; therefore, the challenge of this dataset is the different densities in different directions, which influences the segmentation of the structural segments. In the supervised classification training data, objects of the same category with different density are chosen for comprehensive training. Figure 8 shows some objects with the same category but with different densities or shapes from the training datasets: due to the missing data and differently shaped, the same category could have different structural segment components. Considering the fewer samples of some categories in the dataset, the algorithm took half of the samples of some categories, such as pedestrian, traffic sign and street lamp, as training data, and tested the other remaining half of the samples.

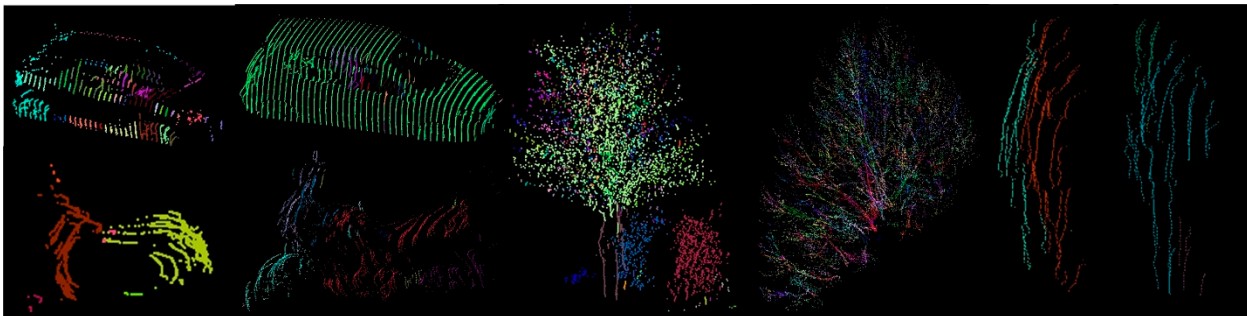

**Figure 8.** Different structural segment components of the same category in the training dataset.

First, the result was qualitatively analyzed by the views of classification. The whole view of the ground-truth of the category is shown in Figure 9a, and the classification results of the proposed method are shown in Figure 9b. It can be seen that most of the objects were recognized correctly by the proposed method, compared with ground-truth data: the main

façade and vegetation achieved the best classification results. The incorrect classifications are shown in Figure 9c—the incorrect points are shown in red. Notably, the incorrect points appeared in small structural segments from the façade's fragment, as show in the green box in Figure 9c. In the yellow box of Figure 9c, because the collected points are too sparse in this region, the quality of the point cloud is too low to retain the shape of the bollard; therefore, misclassification is concentrated in this region.

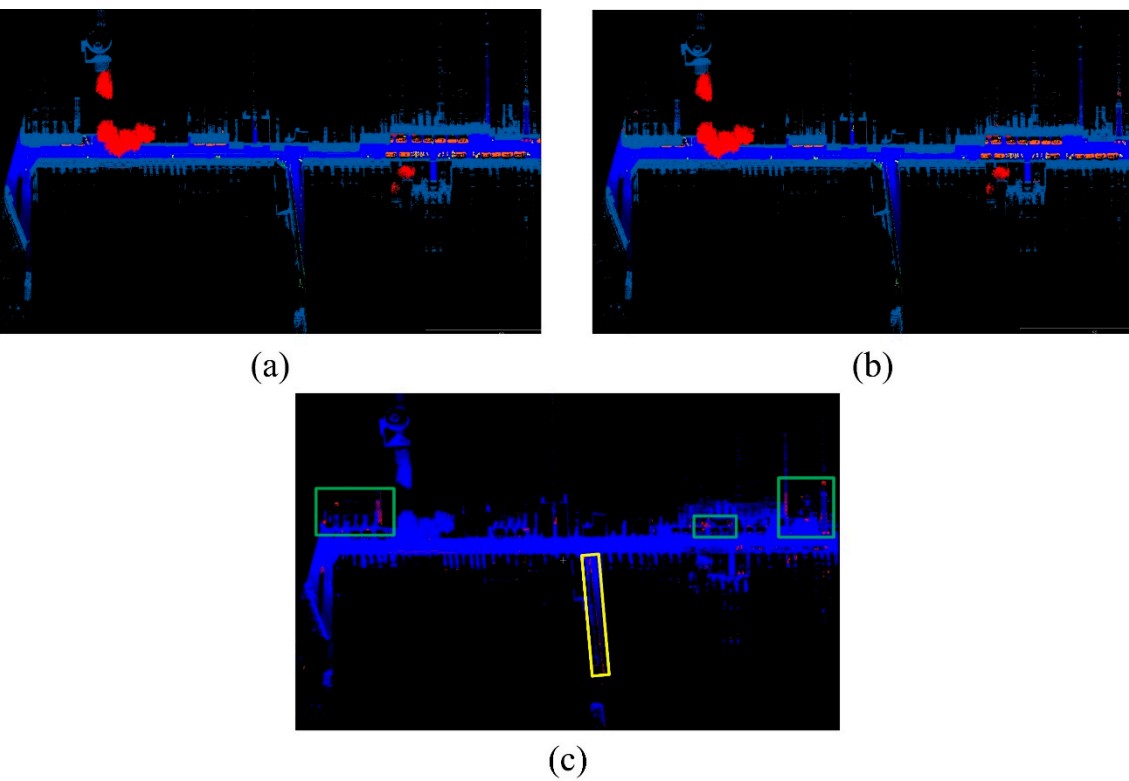

**Figure 9.** Classification in the Paris-rue-Cassette dataset. (**a**) Truth-ground of Paris-rue-Cassette dataset. (**b**) Classification results of the proposed method. (**c**) Incorrect classification.

Figure 10 shows three scenes where incorrect points are concentrated. Figure 10a shows a façade scene where many disperse fragments exist. Due to the discreteness of the fragments, the pairwise term and higher-order term cannot smooth or correct the incorrect classification, which can then easily be mis-identified as other small objects which have similar-shaped features. This situation often occurs in façades because the scanner line generates isolated block point clouds behind the façade, or some irregular objects are attached to the façade. However, it can be seen that the proposed method successfully identified a large number of objects with relatively complete structure, because the method in this paper takes advantage of the structural segments as classification units, which can retain the shape features of objects.

Figure 10b represents an intersection situation between vegetation and façade, with the intersection position marked by a blue box in Figure 4 in Section 2.3.3. It can be seen that the over-segmented structural segments generated by the inverse node graph strategy retained lots of boundaries in Figure 4c, and obtained good classification results even when they were close, as shown in Figure 10b. The main vegetation has been well recognized in Figure 10b. Due to the mixture of walls and vegetation, it is inevitable that a few façade structural segments with vegetation have been classified as vegetation.

Figure 10c shows the classification results of small objects, such as cars, motorcycles and pedestrians. It can be seen that the proposed method has a good ability to recognize cars, but incorrectly recognizes motorcycles as cars or pedestrians because of some similarities between them. As marked in the green box in Figure 10c, some points of the

motorcycles are discontinuous; therefore, parts of the structural segments can easily be incorrectly categorized.

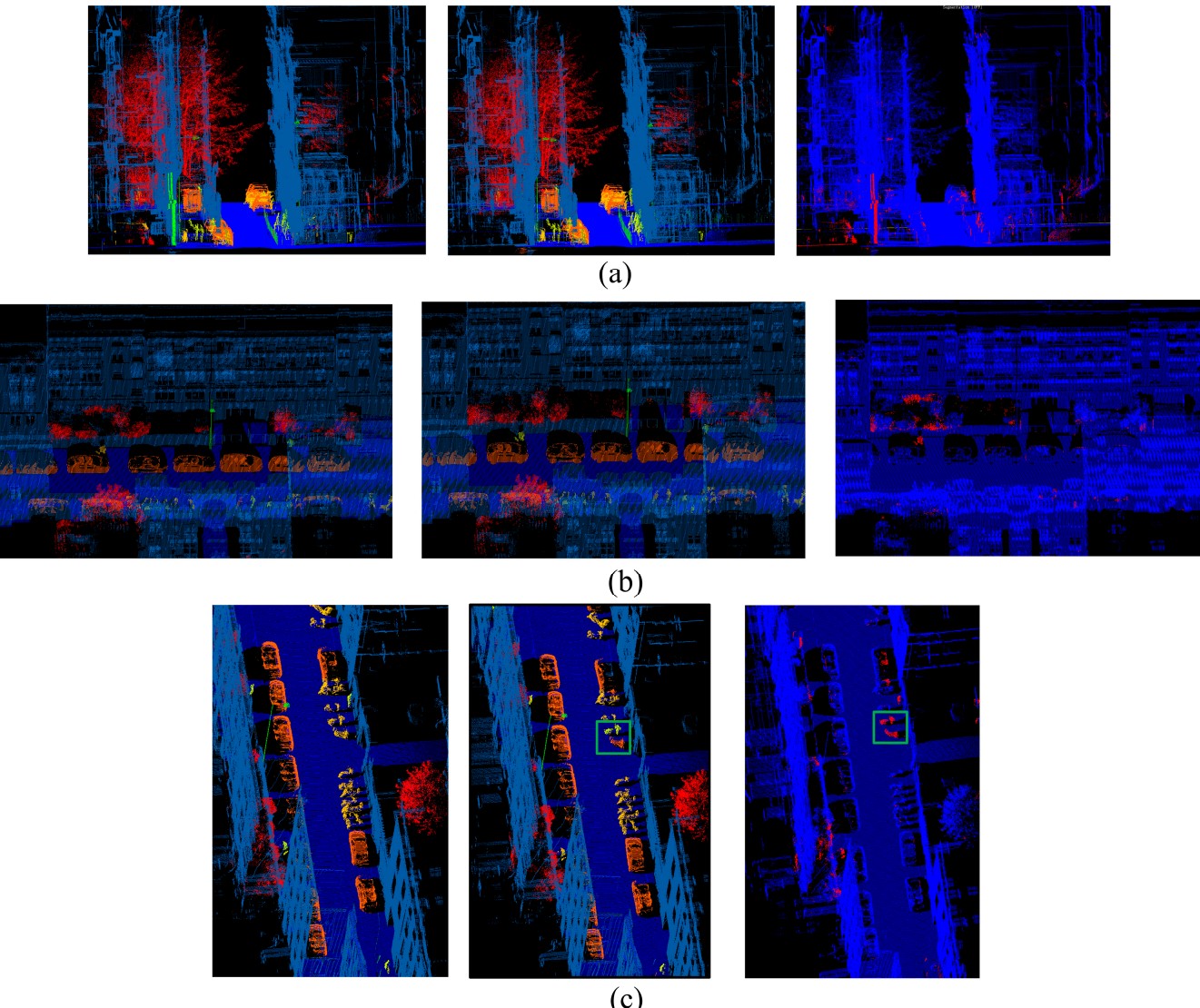

**Figure 10.** Different scene of classification results of the proposed method in the Paris-rue-Cassette dataset, from left to right are the classification results, the truth-ground and the incorrect points. (**a**) Façade scene. (**b**) Intersection objects scene. (**c**) Car, motorcycle and pedestrian scene.

Next, the method in this paper was quantitatively analyzed by basic classification metrics, overall accuracy, recall, precision and F1-score. Three existing semantic classification methods [23,33,40] which were successfully tested on the Paris-rue-Cassette dataset are taken as comparative items. The quantified classification results are shown in Table 4, where the accuracy calculation of the proposed method excluded the training dataset, and the best results in each table are marked in bold. The overall accuracy and mean F1-score of the proposed method were 97.57% and 77.06%, which were 0.44% and 4.86% better than method 3, respectively. Regarding category classifications in Table 5, the proposed method obtained the best recall in the categories façade, vegetation and car, which all have special geometric features, indicating that the proposed method is good for recognizing these common objects in urban scenes. However, for the smaller categories, the proposed method is weaker than others in recognizing motorcycles, pedestrians and traffic signs. One reason is the lack of training data of objects in the small category, which is why the

worst result occurred for traffic signs. Another reason is because the low quality of points generates small fragments in these objects, which may mislead the classifier. Motorcycles and pedestrians have similar geometric features; therefore, it is easy to confuse them, which results in a poor recall result.

**Table 4.** Overall accuracy of different methods in the Paris-rue-Cassette dataset.

|  | Method 1 [23] (%) | Method 2 [40] (%) | Method 3 [33] (%) | Ours (%) |
|---|---|---|---|---|
| Overall Accuracy | 89.60 | 95.74 | 97.13 | **97.57** |
| Mean F1-score | 52.18 | 70.20 | 72.20 | **77.06** |

**Table 5.** Recall of different methods in the Paris-rue-Cassette dataset.

| Recall | Ground (%) | Façade (%) | Vegetation (%) | Car (%) | Motorcycle (%) | Pedestrian (%) | Traffic Sign (%) | Street Lamp (%) | Bollard (%) |
|---|---|---|---|---|---|---|---|---|---|
| Method 1 [23] | 96.46 | 87.21 | 86.02 | 61.12 | 82.85 | 82.25 | 76.57 | - | - |
| Method 2 [40] | 98.22 | 94.21 | 84.78 | 93.07 | **97.58** | **96.87** | 89.63 | - | - |
| Method 3 [33] | **99.45** | 97.95 | 89.59 | 76.56 | 87.86 | 93.36 | **96.16** | 92.74 | **82.64** |
| Ours | 98.87 | **98.45** | **92.06** | **96.12** | 82.74 | 74.33 | 45.36 | **92.97** | 68.15 |

In terms of precision, as shown in Table 6, the proposed method achieved the best performance, which indicates that structural segments segmented from the IN-Graph can retain more typical geometric features for recognizing different categories, reducing the probability of one category being wrongly identified as the other. Although the categories of motorcycle and pedestrian had smaller recall than other methods, they had higher precision than other methods, which means that the proposed method rarely incorrectly recognized other categories as them, but these categories could easily be recognized as other categories. Precision indicates that the proposed method has a good correctness of classification but is not reliable as other methods in recognizing these categories. It infers that due to uneven density, the similar structural segments between different categories can influence the completeness of classification of the proposed method. When the proposed method allocates the correct label as part of structural segments of an object, it can recognize the whole object by higher-order CRF with small-label cluster refinement; therefore, it can improve the precision of identifying objects by recognizing parts of them from the classifier. The F1-scores in Table 7 indicate that the classification results of some categories in the proposed method are superior to other methods when comprehensively considering both the recall and precision.

**Table 6.** Precision of different methods in the Paris-rue-Cassette dataset.

| Precision | Ground (%) | Façade (%) | Vegetation (%) | Car (%) | Motorcycle (%) | Pedestrian (%) | Traffic Sign (%) | Street Lamp (%) | Bollard (%) |
|---|---|---|---|---|---|---|---|---|---|
| Method 1 [23] | **99.24** | 99.24 | 25.66 | 67.67 | 17.74 | 14.95 | 9.24 | - | - |
| Method 2 [40] | 98.71 | 98.71 | 56.62 | 86.08 | 51.99 | 18.99 | 24.88 | - | - |
| Method 3 [33] | 96.89 | 96.89 | **92.43** | 93.07 | 42.46 | 38.52 | 14.63 | 8.36 | 89.63 |
| Ours | 98.46 | **99.32** | 91.32 | **94.94** | **53.98** | **42.08** | **40.54** | **51.66** | **95.43** |

Table 8 demonstrates the efficiency of different classification methods. It can be seen that the pointwise classification [23] takes more time than other methods because it takes points as units, resulting in much more computational complexity. The proposed segment-based classification method is slower than the supervoxel-based classification method [33], because the proposed method involves the generation of multi-scale supervoxels, running the voxelization program multiple times; the proposed method also involves structural segment generation, which takes some time for segmentation. Supervoxel-based classification

does not mention the time cost of higher-order CRF; therefore, the time costs of this step cannot be compared. However, due to the small-label cluster optimization step, the overall time cost of the proposed method will be lower than the supervoxel-based method. It can be concluded that the efficiency of our segment-based method is better than the pointwise method but slower than the supervoxel-based method: the proposed method improves the classification accuracy at the cost of certain efficiency loss.

**Table 7.** F1-score of different methods in the Paris-rue-Cassette dataset.

| F1-Score | Ground (%) | Façade (%) | Vegetation (%) | Car (%) | Motorcycle (%) | Pedestrian (%) | Traffic Sign (%) | Street Lamp (%) | Bollard (%) |
|---|---|---|---|---|---|---|---|---|---|
| Method 1 [23] | 97.83 | 92.85 | 39.53 | 64.23 | 29.23 | 16.61 | 25.01 | - | - |
| Method 2 [40] | 98.47 | 96.85 | 67.90 | 89.43 | **67.84** | 39.60 | 31.34 | - | - |
| Method 3 [33] | 98.15 | 98.60 | 90.99 | 84.01 | 57.25 | **54.54** | 25.40 | 15.34 | **85.99** |
| Ours | **98.66** | **98.88** | **91.69** | **95.54** | 65.33 | 54.74 | **42.81** | 66.41 | 79.52 |

**Table 8.** Time cost of different classification methods in the Paris-rue-Cassette dataset.

| Steps | Pointwise Classification [23] (s) | Supervoxel-Based Classification [33] (s) | Segment-Based Classification (Ours) (s) |
|---|---|---|---|
| Generation of density-consistent regions | - | 76.5 | - |
| Generation of super voxels | - | 142.3 | 294 |
| Generation of structural segments | - | - | 262 |
| Feature computation | 43.47 | 1.9 | 31 |
| Labelling and Clustering | 4100.22 | 0.9 | 9 |
| Higher-order CRF and Refinement | - | - | 491 |
| Total | 4143.69 | 221.6 | 1087 |

### 3.3. Experiment on the Wuhan University TLS Dataset

There were seven categories in the Wuhan University dataset: ground, façade, vegetation, car, motorcycle, pedestrian and artificial pole. As Figure 11 shows, the main category existing in the Wuhan University dataset is vegetation, including a large number of tree crowns and trunks, which makes it a challenge to distinguish between trees and artificial poles. The range of the Wuhan University dataset was large; therefore, the training dataset was obtained from part of the whole dataset, as shown in the red box in Figure 11 and for balancing the composition between different categories, it reduced the amount of vegetation and façades to prevent overfitting. The remaining region of the Wuhan University dataset was taken as the testing dataset, as shown in the blue box in Figure 11.

The classification results of Wuhan University dataset are shown in Figure 12. The first row in Figure 12 is the classification results by the proposed method, the middle row is the ground-truth of the dataset, and the last row is a binary image which shows the correctly classified points (in blue) and incorrectly classified points (in red). Figure 12a is a complete view of the classification results, indicating that the proposed method correctly identified most of the targets in this scene, and worked well in identifying the vegetation, façade and ground, which were among the major components in the dataset. Figure 12b,c show the detailed scenes in which small objects are present; cars, motorcycles and pedestrians were identified well by the proposed method, even though they were close to each other. The results indicate that the proposed method performs well in identifying the objects with distinct structures. The reason is that the proposed method over-segments points into structural segments using IN-Graph, which retains part of the structural features of the objects, and preserves the individuality of single targets. Thus, it can not only identify the target well, but also prevents under-segmentation. As shown in the black box in Figure 13, a

bicycle is parked very close to a tree: Figure 13a shows the segmentation results. It can be observed that even though they were very close to each other, the segmentation by IN-Graph still divided them well. The skeleton of the bicycle was extracted relatively completely, which preserved distinct features to help with classification, as shown in Figure 13b.

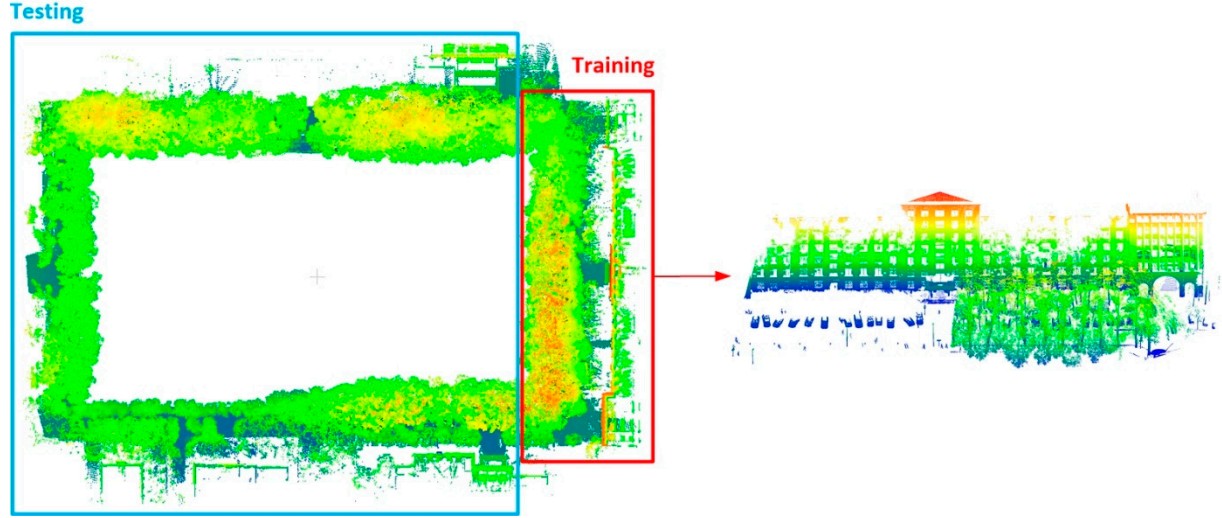

**Figure 11.** Training and testing dataset in the Wuhan University dataset.

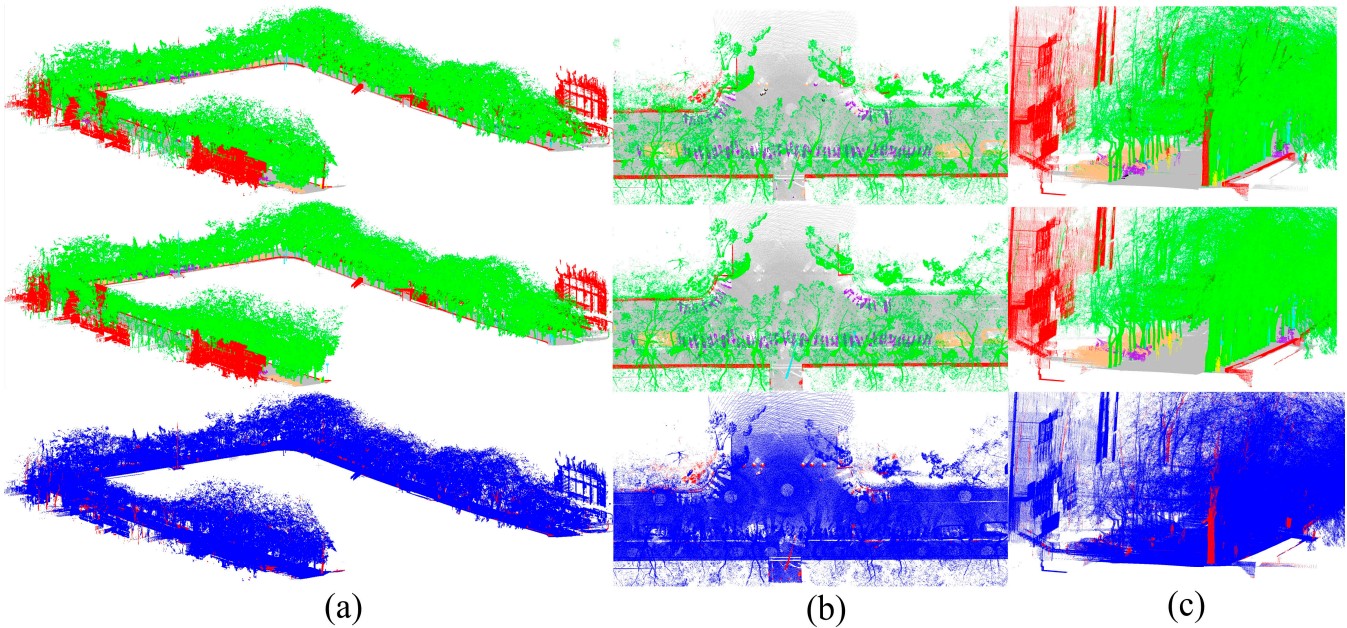

**Figure 12.** Classification results of the proposed method in the Wuhan University dataset. (**a**) Overall. (**b**) and (**c**) Scenes with more classified details.

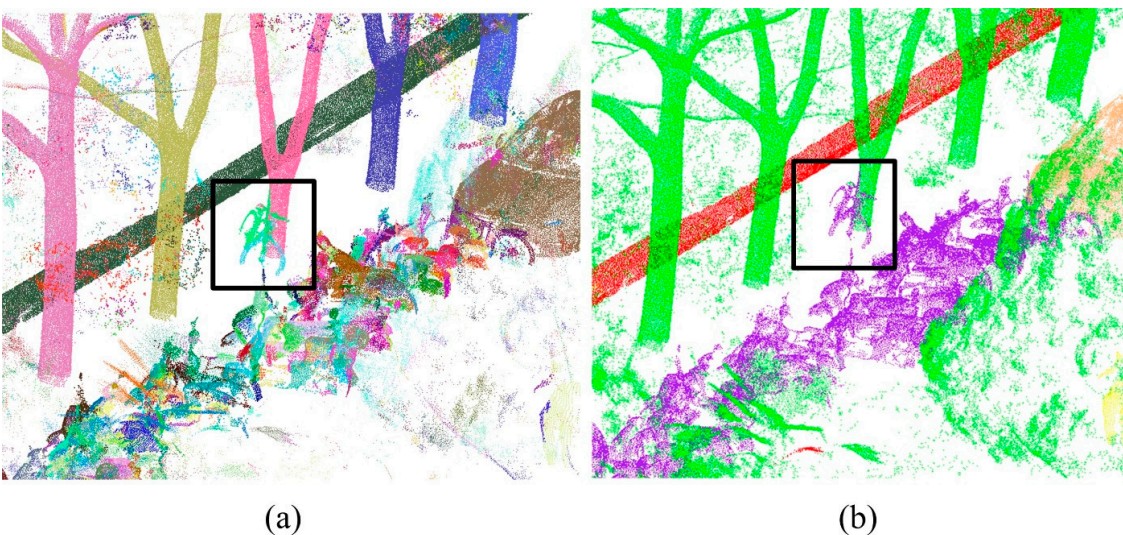

**Figure 13.** The results of the intersecting part processed by the proposed method. (**a**) Segmentation results by IN-Graph. (**b**) Classification results.

In the Wuhan University dataset, there were lots of pole-like objects: these were natural poles such as trees and artificial poles such as lamps and traffic signs. Figure 14 illustrates the recognition capability for distinguishing natural poles and artificial poles. From Figure 14a, it can be seen that most trees have been identified, and artificial poles have also been successfully identified in many regions, as shown in the blue box in Figure 14a. This illustrates that the proposed method has some ability to recognize pole-like objects. However, in Figure 14b,c, misidentification occurred. Figure 14b misidentified a tree as an artificial pole object because the diameter of tree is rather close to the artificial pole-like object. Figure 14c shows a situation where a traffic sign is misidentified as a tree, and the reason could be that the upper-half of this traffic sign stretched to the side, which means it could easily be misidentified as a tree. The examples shown in Figure 14 illustrate that the proposed method has some ability to recognize pole-like objects, but improvements are needed to deal with some special situations.

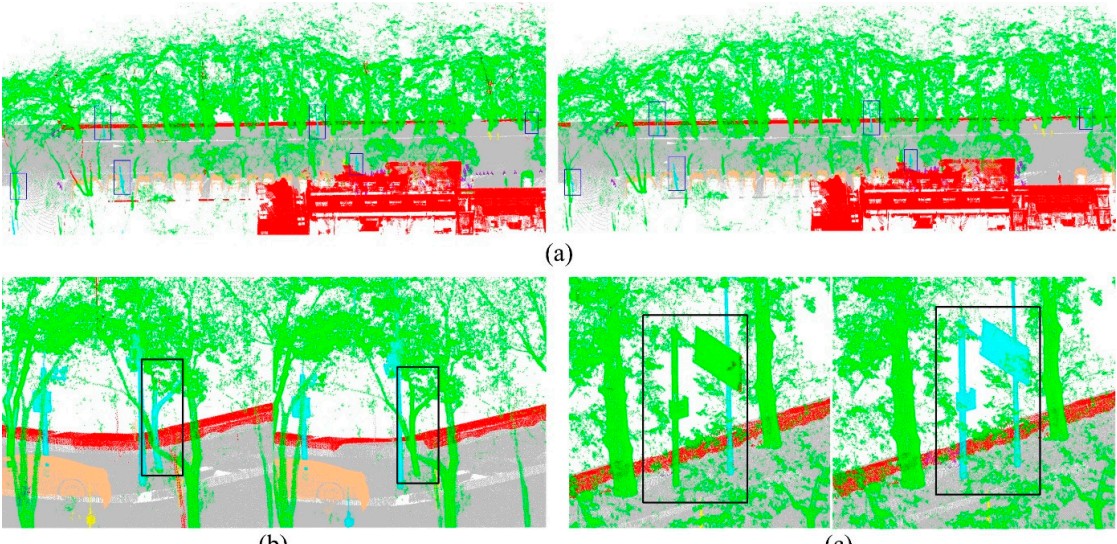

**Figure 14.** Pole-like object identification of the proposed method in the Wuhan University dataset. (**a**) Correct identification of a pole-like object. (**b**,**c**) Misidentification.

Next, the performance of each step in the label refinement phase is discussed. Tables 9–12 quantitatively evaluate the performance of RF, higher-order CRF and higher-order CRF considering small-label clusters; the quantitative evaluation metrics were the same as in Section 3.2. As seen from Table 9, random forest achieved a good overall accuracy of 93.64%, which illustrates that the extracted structural segments provide good materials for classification, indicating the advantages of the segmentation method in this paper and most objects could be directly identified by the classifier; the higher-order CRF improved overall accuracy by 2.02% over the random forest, illustrating that higher-order CRF works well in refining the classification results. The last test was the small-label cluster optimization based on previous steps, which improved overall accuracy by 0.63% and the mean F1-score by 3.26% compared with normal higher-order CRF, indicating that optimization considering small-label clusters can help higher-order CRF to better refine classification results. Figure 15 presents examples illustrating the optimization of each step in the classification. From the orange box in Figure 15a,b, the individual targets have been smoothed by higher-order CRF, and from the blue box in Figure 15b,c, some small, misidentified clusters have been refined by small-label cluster optimization.

**Table 9.** Overall accuracy of the proposed method in different steps.

|  | RF (%) | RF + Higher-Order CRF (%) | Ours (%) |
|---|---|---|---|
| Overall Accuracy | 93.64 | 95.66 | **96.39** |
| Mean F1-score | 80.10 | 82.81 | **86.07** |

**Table 10.** Recall of the proposed method in different steps in the Wuhan University dataset.

| Recall | Ground (%) | Façade (%) | Vegetation (%) | Car (%) | Motorcycle (%) | Pedestrian (%) | Artificial Pole (%) |
|---|---|---|---|---|---|---|---|
| RF | 97.96 | 92.43 | 93.52 | 93.79 | 87.80 | 58.44 | 40.36 |
| RF + Higher-order CRF | 98.18 | 93.04 | 94.77 | 94.24 | 93.01 | 62.07 | 51.00 |
| Ours | **98.23** | **94.83** | **95.93** | **95.02** | **96.79** | **67.82** | **57.78** |

**Table 11.** Precision of the proposed method in different steps in the Wuhan University dataset.

| Precision | Ground (%) | Façade (%) | Vegetation (%) | Car (%) | Motorcycle (%) | Pedestrian (%) | Artificial Pole (%) |
|---|---|---|---|---|---|---|---|
| RF | **99.51** | 73.03 | 98.07 | 77.45 | 73.85 | 84.91 | 66.84 |
| RF + Higher-order CRF | 99.48 | 77.23 | 98.33 | 78.21 | 74.89 | 85.84 | 71.28 |
| Ours | 99.49 | **81.61** | **98.31** | **81.05** | **80.33** | **93.58** | **73.99** |

**Table 12.** F1-score of the proposed method in different steps in the Wuhan University dataset.

| F1-Score | Ground (%) | Façade (%) | Vegetation (%) | Car (%) | Motorcycle (%) | Pedestrian (%) | Artificial Pole (%) |
|---|---|---|---|---|---|---|---|
| RF | 98.73 | 81.60 | 95.74 | 84.84 | 80.22 | 69.23 | 50.33 |
| RF + Higher-order CRF | 98.83 | 84.40 | 96.52 | 85.48 | 82.97 | 72.05 | 59.46 |
| Ours | **98.86** | **87.72** | **97.10** | **87.48** | **87.79** | **78.65** | **64.89** |

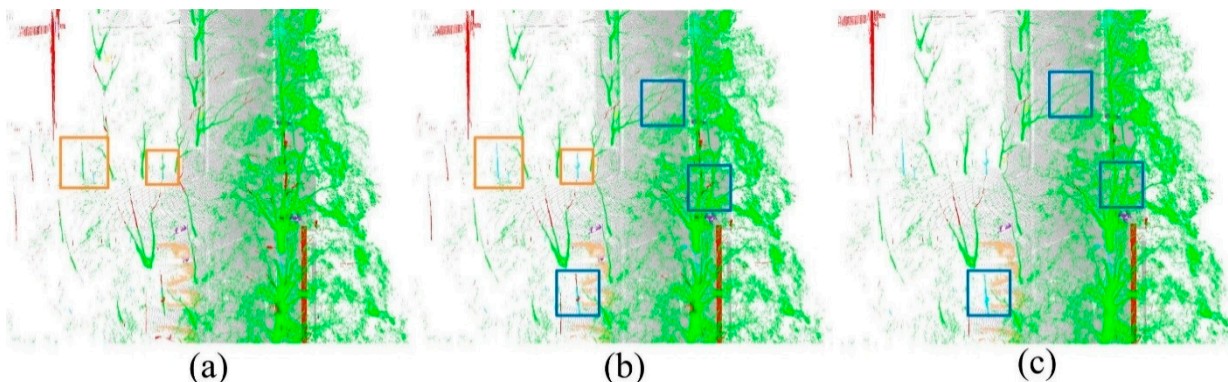

**Figure 15.** Classification results of different steps of the proposed method in the Wuhan University dataset. (**a**) Classification results of RF. (**b**) Classification results of higher-order CRF. (**c**) Classification results of higher-order CRF considering small-label clusters.

Tables 10–12 present more specific evaluations about different categories in recall, precision and F1-score. Table 10 indicates that the proposed method had a good ability to identify the ground, façades, vegetation, cars and motorcycles in the Wuhan University dataset, and it was relatively insensitive to pedestrians and artificial poles. However, it can be seen that the higher-order CRF considering small-label clusters contributed to greatly improving the recall accuracy for pedestrians and artificial poles, achieving improvements of 9.38% for pedestrians and 17.42% for artificial poles over RF, and 5.75% and 6.78% over higher-order CRF, indicating that our refined step can improve the recognition rate of each category. Table 11 presents the precision of each category, and it can be found that pedestrians and artificial poles have better precision than recall, which are 93.58% and 73.99%, respectively, illustrating that the proposed method can identify categories accurately, and the misrecognition of other categories as pedestrians and artificial poles seldom occurs. At the same time, it can be seen that the optimization steps can improve the precision for most targets, and Table 12 shows the same phenomenon in F1-score; the necessity of optimization steps is verified again. Tables 9–12 successfully verify that the proposed method achieved good performance in the Wuhan University dataset, and the contribution of higher-order CRF and further refinements of small-label cluster optimization are also demonstrated.

From Tables 9–12, it can be seen that small-label cluster optimization is useful to refine the label of small clusters in one clique; in theory, it needs to run higher-order CRF for each small cluster to obtain their new labels. This means it refines the label of one small cluster, along with allocating the labels of all segments again, consuming lots of computation time. The simplified algorithm is aimed at the target label cluster, which can reduce the time in label refining more than using higher-order CRF directly. Table 13 provides the time cost of the small-label cluster process in one clique by higher-order CRF and the simplified algorithm; it can be seen that the simplified algorithm can reduce the consumption time.

**Table 13.** Time cost of the small-label cluster refinement process with different algorithms.

| Small-Label Cluster Refinement Algorithm | Time Cost (s) |
| --- | --- |
| Higher-order CRF | 79 |
| Simplified algorithm | 22 |

*3.4. Discussions*

From the experiments on the Paris-rue-Cassette dataset and Wuhan University dataset, the proposed classification method has been evaluated by qualitative and quantitative analysis, which has proven that the proposed method can work for MLS and TLS point clouds. Through the comparison between the different methods, the proposed method obtained the best overall accuracy of 97.57% in the Paris-rue-Cassette dataset, and it had

a better recall in façade, vegetation and car categories, but it achieved lower recall for motorcycles, pedestrians and traffic signs. The overall precision of the proposed method is better than other methods, illustrating that the classification bias is small in the proposed method. Through the results based on the Wuhan University dataset, the contribution of higher-order CRF and small-label optimization in the classification phase have been verified. From Figures 10b and 13, it can be seen that the intersecting objects have been divided, illustrating the advantage of segmentation in the method.

The good performance of the proposed method is attributed to the following two reasons:

- Segmentation by IN-Graph: In our method, segments are taken as the smallest units of classification, because the structural segments can retain the original shape features, which is beneficial for category recognition. The proposed IN-Graph extracts the segments by cutting off the edge between two objects: this is really segmentation rather than clustering. It is not the same as a normal graph which needs to know the properties of a node. The IN-Graph decides the connection state of two objects by exchanging the decision-making position from node to edge without any prior information or definition of the node; therefore, it has more applicability. Additionally, it realizes the optimal segmentation by graph cutting, so the boundaries between objects are retained while preserving as many of the shape features as possible. This is the reason why segmentation by IN-Graph contributes to improving the classification accuracy, as well as its ability to distinguish adjacent objects;

- Higher-order CRF considering small-label clusters: In the refinement step of the classification results, as described above, higher-order CRF works well in small region refinement, but it does not work well in separate clusters in one clique with the same label. In this study, it dealt with these separate clusters with the same label separately, and this paper has proposed a simplified algorithm for a local optimum to allocate labels of these small-label clusters. Thus, it can accurately eliminate small-label clusters in the clique and speed up the operation process. The classification results based on higher-order CRF are further optimized to obtain higher classification accuracy by small-label cluster optimization.

However, some problems need to be fixed in future work. First, comparing the time cost, the efficiency of the proposed method is lower than the supervoxel-based method; the operation time mainly concentrates on the generation of multi-scale supervoxels and structural segments, and small-label cluster optimization also takes some time to refine the classification. This method sacrifices some time efficiency to obtain higher precision units. Secondly, the ability to identify objects with similar shape needs to improve. As shown in Figure 14b,c, the trees and the artificial poles were misidentified, probably due to the fact that the same category has different forms, and the shapes of items in different categories could be very similar, which confuses the classifier. It would be useful to carry out subdivisions of shape features in the future.

## 4. Conclusions

In this paper, a segment-based semantic segmentation method of 3D point clouds is proposed to classify the objects in the urban scene. The novel points of the proposed method can be concluded in two parts: (1) In the segmentation step, an inverse node graph method is proposed to extract the structural segments from multi-scale supervoxels as the basic unit of classification. The IN-Graph has two advantages: ① it does not need any property information of the node in the graph, because the segment is clustered by the connection state of its edge; ② the boundary of intersecting objects is preserved while maintaining the structural segments integrity. (2) In the label refinement step, it individually deals with the small-label cluster based on the higher-order CRF, which further optimizes the mislabeling of the fragments, and a simplified algorithm is carried out to speed up the efficiency of the optimization. The proposed classification method has successfully been tested on the Paris-rue-Cassette MLS dataset and the Wuhan University TLS dataset, in comparison with three other state-of-the-art methods: the method in this paper achieved a better

overall accuracy and mean F1-score, 97.57% and 77.06% in the Paris-rue-Cassette dataset, respectively. Additionally, the optimization step of the proposed method has been verified to be effective through ablation experiments: the results demonstrated that the small-label cluster optimization improved overall accuracy by 2.75% and 0.73%, and the mean F1-score by 5.97% and 3.26%, compared with the RF and higher-order CRF refinement, respectively. Considering each category, the proposed refined step was over 17.42% and 6.78% more precise than random forest and higher-order CRF methods in recognizing artificial poles. From the experiments, the higher-order CRF considering small-label cluster did further improve the classification results.

Although the proposed method has several advantages, it still has some drawbacks which need to be addressed. For instance, it is time-consuming in segment generation and label refinement, so the efficiency is lower than the supervoxel-based method; another example is that the precision of some category is better, but the recall is lower than the state-of-the-art methods. To handle these problems, more simplified segmentation algorithms and valid structural shape features will be investigated in future work.

**Author Contributions:** Conceptualization, B.Z. and X.H. (Xianghong Hua); methodology, B.Z. and K.Y.; data analysis, B.Z., X.H. (Xiaoxing He) and W.X.; data collection, B.Z., Q.L., H.Q. and L.Z.; writing—original draft preparation, B.Z.; writing—review and editing, K.Y. and C.L.; funding acquisition, X.H. (Xianghong Hua). All authors have read and agreed to the published version of the manuscript.

**Funding:** This work was supported in part by the Foundation of Key Laboratory for Digital Land and Resources of Jiangxi Province, East China University of Technology under Grant DLLJ202015, in part by the National Natural Science Foundation of China under Grant 41674005 and Grant 41871373, the China Postdoctoral Science Foundation under Grant 2021T140469 and Grant 2019M663069, and in part by the Guangdong Basic and Applied Basic Research Foundation under Grant 2019A1515111212. Xiaoxing He was funded by a Youth Project of the National Natural Science Foundation of China (Research on Accurate Estimation of Velocity Field on GNSS Reference Stations with Time-varying Signal and Episodic Tremor).

**Institutional Review Board Statement:** Not applicable.

**Informed Consent Statement:** Not applicable.

**Data Availability Statement:** The data presented in this study are available on request from the corresponding author.

**Acknowledgments:** The authors would like to thank the Experimental Teaching Center of Surveying and Mapping, Wuhan University, for the instrument provided for data collection.

**Conflicts of Interest:** The authors declare no conflict of interest.

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
