# Peer review of "An Inverse Node Graph-Based Method for the Urban Scene Segmentation of 3D Point Clouds"

_remotesensing, doi:10.3390/rs13153021_

Round 1

Reviewer 1 Report

The paper presented by the authors shows an inverse node graph based method for urban scene segmentation of 3D point cloud, which is suitable for its publication.

Author Response

Comments and Suggestions for Authors: The paper presented by the authors shows an inverse node graph based method for urban scene segmentation of 3D point cloud, which is suitable for its publication.

Response: Thank you for your review and comments.

Reviewer 2 Report

OK, an interesting paper. row 31 may be, both abbreviations are known, but it will be good to explain them to readers: MLS point dataset and TLS... row 40 and following: for classifying there are also other procedures, it should be cited in the introduction as the other possible solution. For example: 10.14311/AP.2018.58.0165 or doi.org/10.3390/s21072431 row 91 and the following (row 370, etc.) ) clique...what do you mean? Explain it. row 108: The main contributions of this work are summarized as follows: textetext I recommend to make a special heading Project aim (for example). From row 108, it is not a typical text for Introduction. Describe better both laser scanning datasets TLS and MLS (precision, density, instrument, etc.) Why are general traffic signs difficult to find? Table 7 ...Ours 98.66 should be bold? row 724 ...optimization improves - it will be suitable to write some words about this improving - it is really significant improvement??

Author Response

Comments and Suggestions for Authors: OK, an interesting paper. row 31 may be, both abbreviations are known, but it will be good to explain them to readers: MLS point dataset and TLS... row 40 and following: for classifying there are also other procedures, it should be cited in the introduction as the other possible solution. For example: 10.14311/AP.2018.58.0165 or doi.org/10.3390/s21072431 row 91 and the following (row 370, etc.) ) clique...what do you mean? Explain it. row 108: The main contributions of this work are summarized as follows: textetext I recommend to make a special heading Project aim (for example). From row 108, it is not a typical text for Introduction. Describe better both laser scanning datasets TLS and MLS (precision, density, instrument, etc.) Why are general traffic signs difficult to find? Table 7 ...Ours 98.66 should be bold? row 724 ...optimization improves - it will be suitable to write some words about this improving - it is really significant improvement??

Response: Thank you very much for giving us the comments and suggestions to improve our paper. The questions and responses are listed below in red.

Thanks for your comments again.

  1. row 31 may be, both abbreviations are known, but it will be good to explain them to readers: MLS point dataset and TLS...

This mistake has been corrected.

  1. row 40 and following: for classifying there are also other procedures, it should be cited in the introduction as the other possible solution. For example: 10.14311/AP.2018.58.0165 or doi.org/10.3390/s21072431

The introduction has added some relevant works about classifying procedures.

  1. row 91 and the following (row 370, etc.) ) clique...what do you mean? Explain it.

The concept of clique come from undirected graph, means the region where the nodes are connected. Now we explain it as “the clique represents a region where is connected in the graph”, please see the row 366 in the paper.

  1. row 108: The main contributions of this work are summarized as follows: textetext I recommend to make a special heading Project aim (for example). From row 108, it is not a typical text for Introduction.

The format is changed refer to previous papers in the Remote Sensing.

  1. Describe better both laser scanning datasets TLS and MLS (precision, density, instrument, etc.)

The illustration about the datasets is completed.

  1. Why are general traffic signs difficult to find?

Because in Paris-rue-Cassette MLS Dataset, the samples traffic signs are small, where is explained in row 516.

  1. Table 7 ...Ours 98.66 should be bold?

This mistake has been corrected.

  1. row 724 ...optimization improves - it will be suitable to write some words about this improving - it is really significant improvement?

The row 724 I guess is row 714. Now we add some description about this improving. And this part concludes the paper, so we only list the improvement about the whole accuracy. Details can be found in section 2.4.2, where describes more about the improvement of each category, at most in 17.42% and 6.78% than random forest and higher-order CRF in artificial pole. From experiments, the higher-order CRF considering small label cluster did further improve the classification results.

Reviewer 3 Report

The authors have proposed a new segmentation method for classifying objects represented by 3D point clouds and compared it with several other state-of-the-art methods. Overall, I recommend accepting this paper after the following minor errors are fixed:

Lines 154-156: "clothe" (should be "cloth")
Line 178: "29" (I believe it shoud be "[29]")
Line 184: "suepervoxel" (should be "supervoxel")

Author Response

Comments and Suggestions for Authors: The authors have proposed a new segmentation method for classifying objects represented by 3D point clouds and compared it with several other state-of-the-art methods. Overall, I recommend accepting this paper after the following minor errors are fixed:

Lines 154-156: "clothe" (should be "cloth")

Line 178: "29" (I believe it shoud be "[29]")

Line 184: "suepervoxel" (should be "supervoxel")

Response: Thank you for your comments. The errors have been fixed, meanwhile the paper is proofread. Thanks again.
